# Perceive With Confidence: Statistical Safety Assurances for Navigation with Learning-Based Perception

**Anushri Dixit, Zhiting Mei, Meghan Booker, Mariko Storey-Matsutani, Allen Z. Ren, Anirudha Majumdar**

Department of Mechanical and Aerospace Engineering, Princeton University
Email: `anushridixit@ucla.edu`

**Abstract:** Rapid advances in perception have enabled large pre-trained models to be used out of the box for transforming high-dimensional, noisy, and partial observations of the world into rich occupancy representations. However, the reliability of these models and consequently their safe integration onto robots remains unknown when deployed in environments unseen during training. In this work, we address this challenge by rigorously quantifying the uncertainty of pre-trained perception systems for object detection via a novel calibration technique based on conformal prediction. Crucially, this procedure guarantees robustness to distribution shifts in states when perceptual outputs are used in conjunction with a planner. As a result, the calibrated perception system can be used in combination with *any* safe planner to provide an end-to-end statistical assurance on safety in unseen environments. We evaluate the resulting approach, *Perceive with Confidence* (PwC), in simulation and on hardware where a quadruped robot navigates through previously unseen indoor, static environments. These experiments validate the safety assurances for obstacle avoidance provided by PwC and demonstrate up to $40\%$ improvements in empirical safety compared to baselines.

**Keywords:** Uncertainty quantification, occupancy prediction, robot navigation

## 1 Introduction

How can we decide if the outputs of a given perception system are sufficiently reliable for safety-critical robotic tasks such as autonomous navigation? Significant strides in perception over the past few years have enabled large pre-trained models to be used out of the box [1] for tasks such as *occupancy prediction*, which serves as a fundamental building block for navigation. However, current pre-trained models are still not reliable enough for safe integration into many real-world robotic systems. Despite being trained on vast amounts of data, these systems can often fail to generalize to novel environments [2, 3, 4]. In this paper, we ask: *how can we leverage the power of large pre-trained occupancy prediction models while providing safety assurances for robot navigation?*

Consider a legged robot tasked with navigating in a cluttered environment such as a home, office, or warehouse (Figure 1). A typical navigation pipeline for such a system consists of two modules: (i) a perception module that detects obstacles, and (ii) a planner that produces collision-free trajectories assuming accurate perception. However, there are two challenges associated with obtaining reliable outputs from the perception module. First, the environments in which we deploy our robots will be *unseen* during training, and thus require *generalization* to new obstacle geometries, appearances, and other environmental factors. Second, *closed-loop deployment* of the perception system in conjunction with a planner causes a shift in the distribution of *states* (e.g., relative locations to obstacles) that are visited by the robot. Since the robot's planner influences future states, the robot may view obstacles from unfamiliar relative poses (Figure 1) and cause the perception system to fail.

In this paper, we address these challenges by performing rigorous *uncertainty quantification* for the outputs of a pre-trained perception system in order to achieve reliably safe (i.e., collision-free) navigation. We utilize techniques from *conformal prediction* [5] in order to lightly process the outputs of a pre-trained obstacle detection system in a way that provides a *formal assurance* on correctness: with a user-specified probability $1 - \epsilon$, the processed perceptual outputs will correctly detect obstacles in a *new* environment. To enable this, we assume access to a modest-sized (e.g., $|\cdot| =$

8th Conference on Robot Learning (CoRL 2024), Munich, Germany.

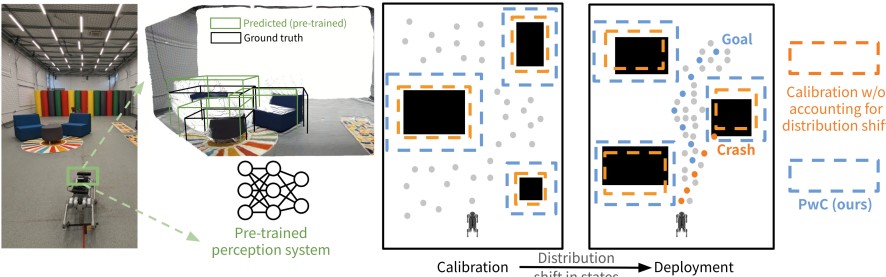

Figure 1: PwC lightly processes the outputs of a pre-trained perception system (green bounding boxes) using conformal prediction in order to ensure a bounded misdetection rate despite *any* distribution shift in states (gray dots). The calibrated perception system (blue boxes) paired with a non-deterministic filter and a safe planner provide an end-to-end statistical assurance on safety in new test environments.

400) dataset of environments that are representative of deployment environments with ground-truth obstacle annotations, and use these for *calibrating* the outputs of the perception system. Crucially, we propose a novel calibration technique that ensures robustness of the perception system to *any closed-loop distribution shift in states*. Hence, the calibrated outputs can be used in conjunction with *any* safe planner to provide an end-to-end statistical assurance on safety in new static environments with a user-specified threshold $1 - \epsilon$. To the best of our knowledge, this is the first work to calibrate a given black-box perception system in a way that ensures robustness to closed-loop distribution shifts in order to provide end-to-end statistical assurances on safe navigation.

Our framework, *Perceive with Confidence* (PwC), is evaluated with experiments in simulation and hardware on the Unitree Go1 quadruped navigating in indoor environments with objects that are unseen during calibration (Figure 1). We validate PwC's ability to provide end-to-end statistical assurances on collision avoidance, while also providing up to $40\%$ increase in safety with only modest reductions in task completion rates compared to baselines that use the pre-trained perception model directly, fine-tune it on the calibration dataset, or utilize conformal prediction for uncertainty quantification but do not account for closed-loop distribution shift.

## 2 Problem Formulation and Overview

**Dynamics and environments.** Suppose that the dynamics of the robot are described by $s_{t+1} = f_E(s_t, a_t)$, where $s_t \in \mathcal{S}$ is the robot's state at time-step $t$, $a_t \in \mathcal{A}$ is the action, and $E \in \mathcal{E}$ is the *environment* that the robot operates in during a given episode. We primarily focus on navigation with static obstacles; in this context, the environment $E$ specifies the locations and geometries of objects. We assume that environments that the robot will be deployed in are drawn from an *unknown* distribution $\mathcal{D}_\mathcal{E}$, e.g., a distribution over possible rooms that the robot may be deployed in. We will make no assumptions on this distribution besides the ability to sample a finite dataset $D = \{E_1, \ldots, E_N\}$ of i.i.d. environments from $\mathcal{D}_\mathcal{E}$.

**Sensor and perception system.** The robot is equipped with a sensor $\sigma : \mathcal{S} \times \mathcal{E} \to \mathcal{O}$ that provides observations $o_t = \sigma(s_t, E)$ (e.g., depth images) based on the robot's state and environment. We assume access to a pre-trained perception model $\phi : \mathcal{O} \to \mathcal{Z}$, which processes raw sensor observations into an occupancy representation of the environment. In this paper, we work with perception models for obstacle detection that output 3D bounding boxes. The representations $(z_0, \ldots, z_t)$ up to the current time-step are aggregated into an overall representation $m_t \in \mathcal{M}$ (e.g., a map).

**Policy.** The representation $m_t$ is used by a planning algorithm in order to produce actions. Denote the resulting end-to-end policy that utilizes a perception model $\phi$ by $\pi^\phi : \mathcal{O}^{t+1} \to \mathcal{Z}^{t+1} \to \mathcal{M} \to \mathcal{A}$, which maps histories of sensor observations to actions.

**Safety and task performance.** Let $C_E^{\text{safe}}$ be a cost function that captures safety (e.g., obstacle avoidance). Specifically, let $\mathcal{S}_{0,E}$ denote the allowable set of initial conditions in environment $E$. Then, $C_E^{\text{safe}}(\pi^\phi) \in \{0, 1\}$ assigns a cost of 0 if policy $\pi^\phi$ maintains safety from any initial state $s_0 \in \mathcal{S}_{0,E}$ when deployed over a given time horizon in environment $E$, and a cost of 1 otherwise. An additional cost function $C_E^{\text{task}}$ can be used to capture task performance (e.g., time to reach a goal).

**Goal: statistical safety assurance.** Our goal is to provide a statistical assurance on safety for the end-to-end policy $\pi^\phi$. We propose a procedure that uses a finite dataset $D$ of environments in order to produce a *calibrated* perception system $\bar{\phi} : \mathcal{O} \xrightarrow{\phi} \mathcal{Z} \xrightarrow{\rho} \mathcal{Z}$. Our approach is modular: outputs of the calibrated perception system may be used with *any* safe planner (cf. Section 5) to ensure:

$$C_{\mathcal{D}_\mathcal{E}}^{\text{safe}}(\pi^{\bar{\phi}}) := \mathbb{E}_{E \sim \mathcal{D}_\mathcal{E}} \left[ C_E^{\text{safe}}(\pi^{\bar{\phi}}) \right] \leq \epsilon, \tag{1}$$

for a user-specified safety tolerance $\epsilon$, while also post-processing outputs from $\phi$ as lightly (i.e., non-conservatively) as possible in order to allow the robot to optimize task performance.

## 3 Background: Conformal Prediction

Conformal prediction (CP) [5, 6] will be our primary tool for performing rigorous uncertainty quantification for perception. Given $N$ i.i.d. (or exchangeable) samples $U_1, \ldots, U_N$ of a scalar random variable $U$, we compute the threshold, $\hat{q}_{1-\epsilon}$, such that the next sample, $U_{\text{test}}$, satisfies,

$$\mathbb{P}[U_{\text{test}} \leq \hat{q}_{1-\epsilon}] \geq 1 - \epsilon, \quad \hat{q}_{1-\epsilon} = \begin{cases} U_{(\lceil (N+1)(1-\epsilon) \rceil)} & \text{if } \lceil (N+1)(1-\epsilon) \rceil \leq N, \\ \infty & \text{otherwise}, \end{cases} \tag{2}$$

where $U_{(1)} \leq U_{(2)} \leq \ldots \leq U_{(N)}$ are the order statistics (sorted values) of the N samples $U_1, \ldots, U_N$. In the CP literature, $U$ is known as the non-conformity score and it is a measure of the (in)correctness of a model. The above guarantee (2) is *marginal*, i.e., (2) holds over the sampling of both the calibration dataset $U_1, \ldots, U_N$ and the test variable $U_{\text{test}}$. Hence, we will need to generate a fresh set of i.i.d. calibration data $\bar{U}_1, \ldots, \bar{U}_N$ for the guarantee to hold for a new sample $\bar{U}_{\text{test}}$. However, in practice, one typically only has access to a single dataset of examples; inferences from this dataset must be used for all future predictions on test examples. In this work, we use the following dataset-conditional guarantee [7, 8] that doesn't require us to generate of $N$ new samples for every test prediction and holds with probability $1 - \delta$ over the sampling of the calibration dataset:

$$\mathbb{P}[U_{\text{test}} \leq \hat{q}_{1-\epsilon} | U_1, \ldots, U_N] \geq \text{Beta}_{N+1-v,v}^{-1}(\delta), \quad v := \lfloor (N+1)\hat{\epsilon} \rfloor, \tag{3}$$

where, $\text{Beta}_{N+1-v,v}^{-1}(\delta)$ is the $\delta-$quantile of the Beta distribution with parameters $N + 1 - v$ and $v$, and we can choose $\hat{\epsilon}$ to achieve the desired $1 - \epsilon$ coverage.

## 4 Offline: Calibrating the Perception System

In this section, we describe our approach to the uncertainty quantification of a pre-trained perception system. We focus on the challenges highlighted in Section 1: providing statistical assurances on safe generalization to novel environments and ensuring that the offline calibration procedure is robust to shifts in the distribution of states induced by the online implementation of the planner.

### 4.1 Misdetection Rate

We focus on perception systems that output bounding boxes that predict the locations of objects in the environment. As an example, Figure 1 (left) shows one such real-world environment wherein the union $A$ of the black boxes denotes the ground-truth locations of the chairs. Let $B_s$ denote the union of the green bounding boxes predicted by the perception system $\phi$ from robot state $s \in \mathcal{S}$. Since the environment in which the robot is deployed may contain partially occluded objects that $\phi$ was not explicitly trained on, the perception system's outputs may be inaccurate.

Our key idea for ensuring *generalization* to new, unseen environments and tackling *the distribution shift arising from the closed-loop deployment* of the calibrated perception system with a plannner is to use a *policy-independent* misdetection cost, $\bar{C}_E$, which considers worst-case errors across *all* states in an environment[1], $\bar{C}_E(\phi) := \max_{s \in \mathcal{S}} \mathbb{1}_{A \not\subseteq B_s}$. We will present a calibration procedure that bounds this misdetection cost with high probability in a new environment, and thus guarantee the correctness of the calibrated perception system independent of the robot policy using CP.

---

[1]It would be infeasible to consider *all* possible states in an environment. In practice, we use a sampling-based motion planner and consider a fixed set of samples for our calibration that could be used by any planner.

### 4.2 Calibration Procedure

**Dataset.** We assume access to a dataset of $N$ i.i.d. environments $D = \{E_1, \ldots E_N\} \sim \mathcal{D}_{\mathcal{E}}$ (cf. Section 2). In each environment, $E_i$, we have access to the union $A_i$ of the ground-truth bounding boxes of all the objects in the environment and the unions $B_{s,i}$ of the predicted bounding boxes generated by the pre-trained perception system $\phi$ from each state $s \in \mathcal{S}$. Care is required to ensure that the calibration environments are representative of deployment environments. As such, we construct the calibration dataset either using real-world environments or create simulation environments using real-world data [9, 10, 11] to ensure sufficient variation in environmental factors (e.g., geometry and locations of obstacles, lighting, etc.).

**Calibration.** In each calibration environment $E_i$, we find the inflation $\Delta_{q_i}$ of the bounding box predictions $B_{s,i}$ so as to ensure that all the ground-truth boxes are fully enclosed by the inflated boxes, i.e, $A \subseteq B_{s,i} + \Delta_{q_i}, \forall s \in \mathcal{S}$, where, $\mathcal{S}$ is assumed to be a finite, discrete set. Here, $B_{s,i} + \Delta_{q_i}$ is the inflation of each bounding box in the union $B_{s,i}$ by $2q_i$ along each dimension. We define the *non-conformity score* for environment $E_i$ to be the minimum required inflation in that environment:

$$U_i = \min_{q_i} \quad q_i \quad \text{s.t} \quad A_i \subseteq B_{s,i} + \Delta_{q_i}, \forall s \in \mathcal{S}. \tag{4}$$

Observe that $U_i \leq 0 \implies A_i \subseteq B_{s,i}, \forall s \in \mathcal{S}$ and a growing $U_i$ signals a worse performance of the pre-trained perception system. We can compute the nonconformity scores for the i.i.d. sampled environments $\{E_1, \ldots, E_N\}$ and the quantile $\hat{q}_{1-\epsilon} = \text{Quantile}\left(U_{(1)}, \ldots, U_{(N)}; \frac{\lceil(N+1)(1-\hat{\epsilon})\rceil}{N}\right)$ [2].

**Proposition 1** *Consider the calibrated perception system $\bar{\phi}$ that modifies every bounding box output of the perception system $\phi$ by scaling the predicted bounding boxes as $\bar{B} = B + \Delta_{\hat{q}_{1-\epsilon}}$. With probability $1 - \delta$ over the sampling of the dataset used for calibration, the calibrated perception system, $\bar{\phi}$, is guaranteed to have an $\epsilon$-bounded misdetection rate on* new *test environments:*

$$\mathbb{E}_{E_{test} \sim \mathcal{D}_{\mathcal{E}}}\left[\bar{C}_{E_{test}}(\bar{\phi})|U_1, \ldots, U_N\right] \leq \epsilon. \tag{5}$$

The above proposition (proof in Appendix A) gives us a formal assurance on the correctness of the perception system *independent of the robot's policy*. As we describe below, the calibrated perception can thus be combined with *any* safe planner to bound the collision rate to $\epsilon$.

## 5 Online: Perception and Planning

We now focus on the online implementation of the method described in Section 4 to reduce conservatism when used in conjunction with a safe planner. In general, a safe planner takes into account the dynamics of the robot and produces plans in the state space $\mathcal{S}$. We call $\mathcal{X}$ the configuration space of the robot (e.g., $x$-$y$ location for a point). For any given environment $E$, we partition $\mathcal{X}$ into three sub-spaces: the known free space $\mathcal{X}^{\text{free}}$, known occupied space $\mathcal{X}^{\text{occ}}$, and unknown space $\mathcal{X}^{\text{unknown}}$.

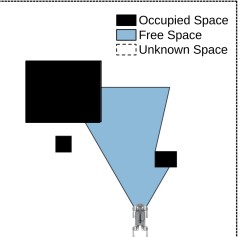
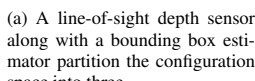
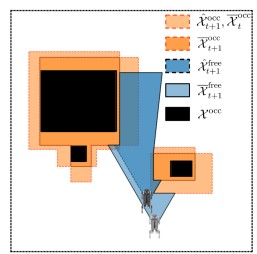

(a) A line-of-sight depth sensor along with a bounding box estimator partition the configuration space into three.

(b) The non-deterministic filter takes intersection over the occupied space and takes union over the free space.

Figure 2

**Non-deterministic filter.** We utilize the assurance obtained from Section 4 to implement a *non-deterministic filter* [12, Ch. 11.2.2] which shrinks the occupied space and grows the known free space over time. Suppose the robot's perceived partition of the state space $\mathcal{X}$ at time $t$ is denoted by the triplet $\{\overline{\mathcal{X}}_t^{\text{free}}, \overline{\mathcal{X}}_t^{\text{occ}}, \overline{\mathcal{X}}_t^{\text{unknown}}\}$, which represents the overall map $m_t$ of the environment. At a new time step $t + 1$, the robot's perception system returns a new estimation for the occupied space, $\hat{\mathcal{X}}_{t+1}^{\text{occ}}$. The filter intersects the occupied spaces: $\overline{\mathcal{X}}_{t+1}^{\text{occ}} = \overline{\mathcal{X}}_t^{\text{occ}} \cap \hat{\mathcal{X}}_{t+1}^{\text{occ}}$. We compute the new estimation of free space $\hat{\mathcal{X}}_{t+1}^{\text{free}}$ based on $\overline{\mathcal{X}}_{t+1}^{\text{occ}}$, considering occlusion and limited field of view.

---

[2]$\hat{\epsilon}$ is the calibration threshold such that the dataset conditional guarantee (3) achieves the desired $(1 - \epsilon)$−coverage with probability $1 - \delta = 0.99$ over the sampling of the calibration dataset.

The new perceived free space is updated by taking the union: $\overline{\mathcal{X}}_{t+1}^{\text{free}} = \overline{\mathcal{X}}_t^{\text{free}} \cup \hat{\mathcal{X}}_{t+1}^{\text{free}}$. The non-deterministic filter pairs effectively with our method in Section 4 for two key reasons: 1) it mitigates the conservatism of our bounding box expansion by intersecting $\overline{\mathcal{X}}_t^{\text{occ}}$, rapidly reducing its size even if the initial prediction with CP bounds appears generous; and 2) Prop. 1 ensures that with high probability in a new test environment, $\overline{\mathcal{X}}_t^{\text{free}}$ never intersects the true occupied space $\mathcal{X}^{\text{occ}}$. We demonstrate the rapid expansion of known free space in Figure 3 for our simulated setup (Sec. 6).

**Safe planning.** With our formal assurance on the estimated free space $\overline{\mathcal{X}}_t^{\text{free}}$, we can utilize *any* safe planner [13, 14, 15] to ensure end-to-end safety, as long as the planner includes a safety filter that takes into account the robot's dynamics in order to reject potentially unsafe actions with the assumption of known state and static (but unknown) environment [16, Corollary 1.4]. For our simulation and hardware experiments, we use the safe planner proposed in [17], which enforces an inevitable collision set (ICS) constraint [18]. We describe implementation details in Appendix C.

**Proposition 2** *For any user-specified safety tolerance $\epsilon$, the* calibrated *perception system $\bar{\phi}$ in Proposition 1 combined with any safe planner that chooses actions based on the outputs of the non-deterministic filter ensures the end-to-end safety for the overall policy $\pi^{\bar{\phi}}$:*

$$C_{\mathcal{D}_{\mathcal{E}}}^{safe}(\pi^{\bar{\phi}}) := \mathbb{E}_{E \sim \mathcal{D}_{\mathcal{E}}} \left[ C_E^{safe}(\pi^{\bar{\phi}}) \right] \leq \epsilon, \tag{6}$$

*where $C_E^{safe}(\pi^{\bar{\phi}})$ is the cost function for safety from Section 2.*

This result (proved in Appendix D) is a direct consequence of the formal assurance on the calibrated perception system that ensures correctness from *any* state in a new test environment (sampled i.i.d. from the same distribution as the calibration environments) with probability $1-\epsilon$ *over environments*.

## 6  Simulated Experiments: Vision-Based Navigation

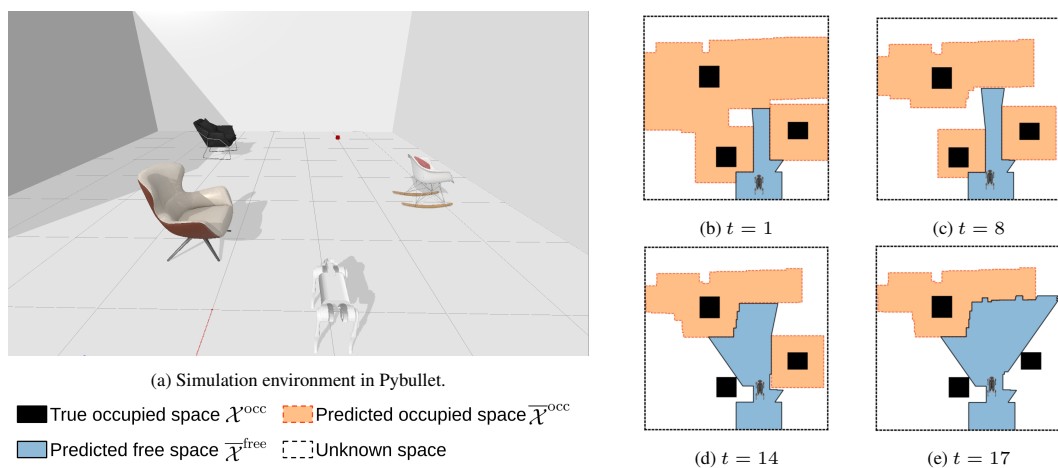

(a) Simulation environment in Pybullet.

■ True occupied space $\mathcal{X}^{\text{occ}}$   ▨ Predicted occupied space $\overline{\mathcal{X}}^{\text{occ}}$

▨ Predicted free space $\overline{\mathcal{X}}^{\text{free}}$   ⬚ Unknown space

(b) $t = 1$     (c) $t = 8$     (d) $t = 14$     (e) $t = 17$

Figure 3: Simulation and non-deterministic filter updates. **(a)** An example environment in simulation. **(b - d)** The robot begins with large occupied space predictions due to the inflation obtained through offline calibration (Section 4). After a few updates, the predicted occupied space $\overline{\mathcal{X}}^{\text{occ}}$ shrinks significantly.

We evaluate our approach for vision-based navigation in the PyBullet simulator [19] using a diverse set of chairs from the 3D-Front dataset [11]. We use the 3DETR end-to-end transformer model [20] as our pre-trained perception system.

**Baselines.** We compare our approach (*Perceive with Confidence* — PwC) to three baselines to illustrate its effectiveness in achieving a user-specified safety rate. First, we consider the most common approach of directly using the outputs of the perception system [20] in our planning pipeline. We call this baseline **3DETR**. Next, we consider the common practice of fine-tuning the outputs of the perception system using a small dataset of task-representative environments $D_{\text{tune}}$ (cf. Section E.1). We call this perception system **3DETR-fine-tuned**. Lastly, we perform calibration using conformal prediction; however, instead of accounting for the closed-loop distribution shift, we bound the

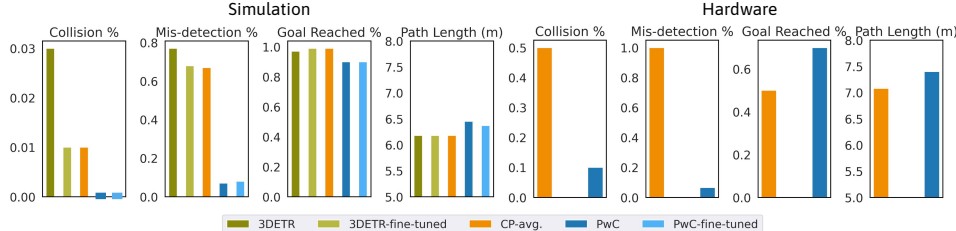

Figure 4: **(Left)** Results for the simulated experiments described in Section 6. Simulations are across 100 new environments with 1 - 5 chairs. **(Right)** Results for the hardware trials described in Section 7. Experiments are across 30 different chair configurations with 4-8 chairs present in each configuration. Here the path length is averaged only for successful trials for both PwC and CP-avg. due to the varying goal locations.

misdetection rate averaged across environments *and* states (similar to [21], which does not utilize conformal prediction, but quantifies expected errors in a perception system for a pre-defined distribution of states). We refer to this baseline as **CP- avg**. We consider two variations of our approach for comparison to the above baselines. First, we refine 3DETR outputs using our calibration procedure described in Section 4. We call this approach **PwC**. Second, the 3DETR outputs are fine-tuned and calibrated using split conformal prediction as described in Appendix E.1; we call this approach **PwC-fine-tuned**. Details regarding calibration and the planner setup are provided in Appendix F.

**Results: Misdetection Rate.** We first compare our method, PwC, to the baseline CP-avg that is also calibrated using conformal prediction but without accounting for the closed-loop distribution shift. We compare the misdetection rate, i.e., whether obstacles in the scene are classified as free space at any point during a trial. We vary the allowable misdetection bound $\epsilon$ for each method, and compute the rate of misdetections in 100 test environments. As seen in Figure 5, our method is guarantees a rate of misdetection lower than the threshold $\epsilon$ while CP-avg violates this threshold for every $\epsilon$ considered.

**Results: Collision Rate.** We compare PwC to the baselines in 100 new environments drawn from the same distribution as calibration environments. Figure 3 illustrates one such test environment and the evolution of the free space in this environment using PwC. Figure 3 shows that though the initial calibrated perception system outputs are inflated, the non-deterministic filter is able to expand the predicted free space in a few time steps and ensure that the robot can navigate without unnecessary conservatism. The results are summarized in Figure 4 and the metrics for success and failure are described in Appendix F. We observe that our proposed approaches, PwC and PwC-fine-tuned, have no collisions in any environments. While the robot reaches the goal in a slightly lower percentage of environments compared to baselines, we emphasize that ours is the only approach that is able to ensure a low, statistically guaranteed misdetection rate across test environments.

Figure 5: As we relax the confidence threshold by increasing $\epsilon$, the misdetection rate increases but remains bounded for PwC. The baseline method has a misdetection rate much higher than acceptable.

To further illustrate the effect of misdetections on safety, we consider a different distribution of environments wherein we randomly place a *single* chair in the straight line path between the initial position of the robot and the goal. For a safety threshold $1 - \epsilon = 0.85$, we compare PwC, CP-avg, and 3DETR. The results are provided in Figure 6 for 100 new test environments, wherein the goal is reached if the robot navigates to within 2 m of the goal. In these environments, the desired safety rate is not met by the baselines while our approach is still statistically guaranteed to be safe.

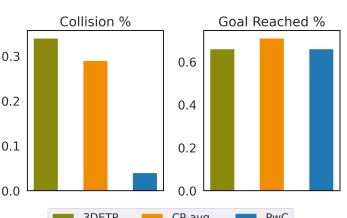

Figure 6: A comparison between collision rates of different perception systems that use the same planner.

We provide additional simulation results to illustrate the effects of 1) closed-loop distribution shifts on safety in Appendix F.4 wherein PwC is robust to an increase in the level of closed-loop distribution shift while the baseline, CP-avg., is not which leads to higher collision rates for CP-avg., 2) the

tradeoff in different partition sizes for fine-tuning using split-CP in Appendix E.1.2, 3) the effect of varying $\epsilon$ on the safety rate in Appendix F.2, 4) impact of using different number of sampled configurations for calibration and online planning in Appendix F.3, and 5) comparison against additional uncertainty-aware perception systems that use a heuristic notion of uncertainty in Appendix F.5.

## 7 Hardware Validation: Vision-Based Quadruped Navigation

We now validate the end-to-end statistical safety assurance of our approach on a quadrupedal hardware platform. As in our simulation setup in Section 6, the robot is tasked with navigating to a goal location while avoiding different chairs placed in varying configurations across a 8m x 8m room. We utilize the perception system calibrated in simulation with a guaranteed safety rate of $1 - \epsilon = 0.85$, and compare our PwC method against CP-avg. (defined in Section 6) across 30 different physical environments (60 trials total). One challenge is to ensure a minimal sim-to-real gap for perception. In order to address this, we utilize depth measurements as the robot's sensory input. This choice facilitates a small sim-to-real gap, as observed in prior work [22, 23]. See Appendix G for more details about the hardware setup.

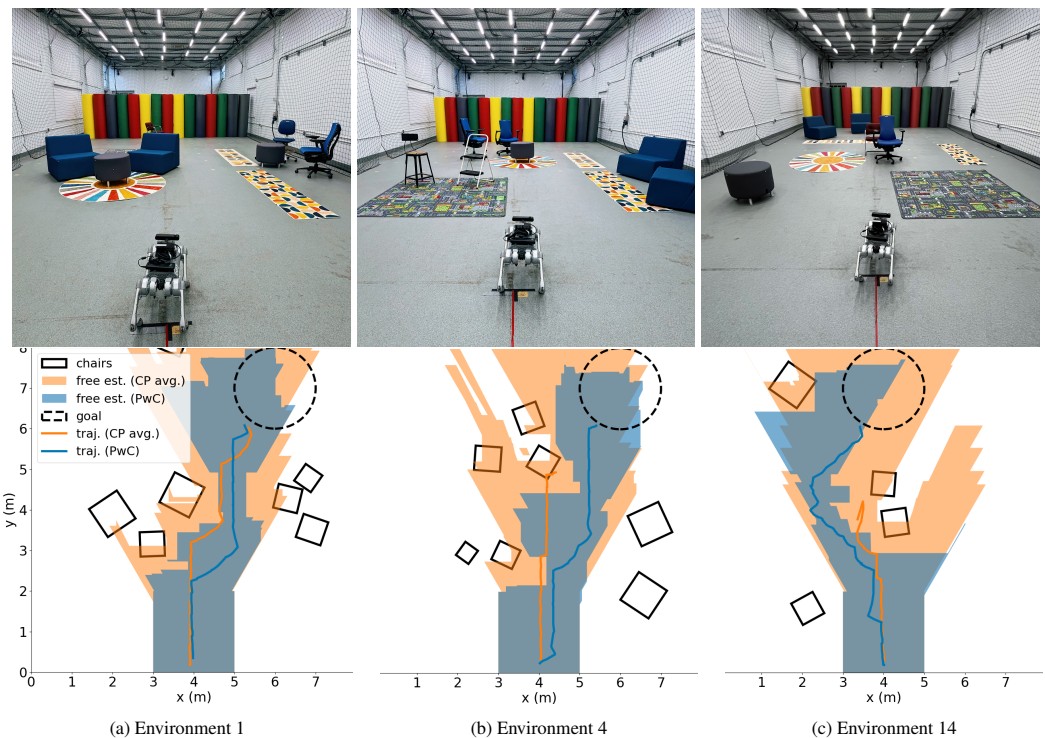

(a) Environment 1          (b) Environment 4          (c) Environment 14

Figure 7: **(Top)** The physical layouts of the example hardware trails. **(Bottom)** A bird's-eye view of the estimated free spaces (shaded regions), and the trajectories performed by the robot (solid lines) with our method (blue) and the baseline (orange). In all three trials, PwC is able to successfully navigate to the goal through narrow paths (in Environment 1) and occluded areas/goal (in Environment 3). Baseline approach, CP-avg., misdetects free space in all environments leading to collisions in Environments 2 and 3.

**Results.** For PwC, we used the $\hat{q}_{0.85} = 0.73$m threshold found in simulation to inflate the predicted bounding boxes returned from 3DETR in order to achieve 85% confidence that our robot will remain safe in new environments. We summarize key statistics of PwC compared against CP-avg. ($\hat{q}_{0.85} = 0.02$) across 30 different environments in Figure 4 (right). Importantly, our trials demonstrate that our confidence bound holds on hardware in real environments and without being too conservative. PwC was safe through 90% of the trials and also had comparable path length to the baseline. Meanwhile, the baseline struggled in the real environments by having misdetections in each trial and colliding with a chair in half of the trials. See Figure 7 for trajectories and free space estimations through several environments with narrow spaces, occluded chairs, and occluded goals. The supplementary video contains full example trials.

PwC's low misdetection rate and higher success rate in these trials emphasize the efficacy of the bounding box inflation provided by CP paired with the non-deterministic filter. This pairing, in a principled way, inflates the (potentially poor) bounding box detections to properly capture obstacles but quickly shrinks the occupied space with the filter such that the robot can still navigate effectively.

## 8 Related Work

**Safe planning.** Collision avoidance is a crucial goal in autonomous navigation. Safe planning methods typically rely on the assumption that the robot has perfect knowledge of its state and environment [16]. Recent approaches have allowed for occlusion [17, 24, 25, 26] or accounted for losing sight of a previously tracked object [27], but still require either perfect detection of seen objects or bounded sensor noise. Such assumptions are impractical for learning-based perception modules that can fail catastrophically in new environments.

**Formal assurances for perception-based control.** Proposed methods include control barrier functions (CBFs) [28, 29], verification methods on neural networks (NNs) [30, 31], and other learning-based methods [31, 32, 33, 21, 34, 35, 36, 37]. However, these works either do not guarantee generalization to novel environments [30, 31], or ignore closed-loop distribution shifts [34, 21], or require end-to-end training and a good prior [35, 36, 37], or demand usage/design of specific controllers [28, 29, 32, 38]. Some make strong assumptions on the perception system [39, 40] that are unrealistic for deployment. In contrast, our method doesn't need any of the above, and is lightweight and modular, allowing for the use of any downstream safe planners to ensure end-to-end safety.

**Conformal prediction.** Conformal prediction (CP) [5, 7, 8] is an uncertainty quantification framework particularly suitable for robotics applications [41, 42, 43, 44] where learned modules are deployed in environments drawn form unknown distributions. In this work, we focus on providing uncertainty quantification for the perception system, which usually involves high-dimensional inputs and closed-loop distribution shifts. Prior works [43, 21, 45, 46] either provide guarantees for a single environment, assume known environments, or do not account for closed-loop distribution shifts. To the best of our knowledge, this is the first work to obtain end-to-end safety assurances for the perception and planning system in new environments while being robust to closed-loop distribution shifts and amenable to changes in the planner parameters.

## 9 Discussion and Conclusions

We presented a modular framework for rigorously quantifying the uncertainty of a pre-trained perception model in order to provide an end-to-end statistical safety assurance for perception-based navigation tasks. Notably, our statistical assurance holds for generalization to new environmental factors (e.g, new obstacle geometries and configurations) and allows for the distribution shift of states that may occur during closed-loop deployment of the perception system with the planner. Our simulation and hardware experiments validated the theoretical safety assurances provided by PwC, while demonstrating significant empirical improvements in safety compared to baseline approaches that do not consider closed-loop distribution shift.

**Limitations and future work.** One limitation of our work is the assumption of static obstacles. As a future direction, we are interested in quantifying uncertainty in both the state of agents moving in the environment and predictions of their *semantic labels* (e.g., "pedestrian" vs. "bicyclist"), and utilizing game-theoretic planning techniques that account for the uncertainty in the agents' current state and future motion. Additionally, the inflation of bounding boxes we acquire from CP introduces some conservatism. We outline an extension to our approach in Appendix E to address this challenge by utilizing more general occupancy representations beyond bounding boxes, e.g., scene completion networks [47], which produce voxel-wise occupancy confidences. Constructing different non-conformity score functions that incorporate confidences from a pre-trained model could also potentially reduce conservatism. Lastly, we are interested in uncertainty quantification for perception models that support tasks beyond point-to-point navigation, e.g., calibrating the outputs of multi-modal foundation models for language-instructed navigation where we ensure accurate detection. We expect that rigorous uncertainty quantification is a necessary step towards fully leveraging the power of large foundation models [1] while safely integrating them into future robotic systems.

## Acknowledgments

The authors were partially supported by the Toyota Research Institute (TRI), the NSF CAREER Award [#2044149], the Office of Naval Research [N00014-23-1-2148], and the Princeton SEAS Innovation Award from The Addy Fund for Excellence in Engineering. This article solely reflects the opinions and conclusions of its authors and not NSF, ONR, TRI or any other Toyota entity.

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

# A Proof of Proposition 1

As seen in Appendix 3, conformal prediction gives us the following *dataset-conditional* guarantee on a new sample of the nonconformity score $U_{\text{test}}$ corresponding to a test environment $E_{\text{test}}$. With probability $1 - \delta$ over the sampling of $U_1, \ldots, U_N$,

$$\mathbb{P}[U_{\text{test}} \leq \hat{q}_{1-\epsilon} | U_1, \ldots, U_N] \geq \text{Beta}_{N+1-v,v}^{-1}(\delta).$$

We can rewrite the event $U_{\text{test}} \leq \hat{q}_{1-\epsilon}$ as:

$$\{U_{\text{test}} \leq \hat{q}_{1-\epsilon}\}$$
$$= \left\{ \hat{q}_{1-\epsilon} \geq \min_{q_{\text{test}}} \ q_{\text{test}} | A_{\text{test}} \subseteq B_{s,\text{test}} + \Delta_{q_{\text{test}}}, \forall s \in \mathcal{S} \right\}$$
$$= \left\{ A_{\text{test}} \subseteq B_{s,\text{test}} + \Delta_{\hat{q}_{1-\epsilon}}, \forall s \in \mathcal{S} \right\}$$
$$= \{A_{\text{test}} \subseteq \bar{B}_{s,\text{test}}, \forall s \in \mathcal{S}\}$$
$$= \left\{ \bar{C}_{E_{\text{test}}}(\bar{\phi}) = 0 \right\},$$

which gives us the desired result (5).

# B Implementation with a limited field-of-view

A natural question that arises after following the calibration procedure described above is: what happens if the robot is not able to observe all objects in the environment from all states? This may happen due to a limited sensing capability or because some parts of the environment are occluded from view. We address this issue in our calibration procedure implementation by only taking into account perception errors for objects that are within the field-of-view of the robot in a given state, and masking any ground-truth bounding boxes that are not visible to the robot, i.e., $A$ (which now depends on state $s$) is the union of all the ground-truth bounding boxes of the *visible* objects. Hence, the perception system correctness assurance stated above holds for all objects within the field-of-view of the robot at any given state. The presence of possibly occluded obstacles are dealt with by a safe planner, which we describe next.

# C Planner implementation details

For our simulation and hardware experiments, we use the safe planner proposed in [17] due to its approximate optimality and ease of implementation. The safety filter in this case is an inevitable collision set (ICS) constraint [18], where the robot is forbidden to enter any state that will eventually result in collision no matter what control actions are taken. Within the known free space $\overline{\mathcal{X}}_t^{\text{free}}$, the robot plans using the fast marching tree algorithm (FMT$^\star$) [48] with dynamics [49]. If the goal is not visible within $\overline{\mathcal{X}}_t^{\text{free}}$, the robot plans to an intermediate goal on the boundary of its free space. The intermediate goals are chosen based on the cost-to-come from current robot state to the intermediate goal, and the distance-to-go from the intermediate goal to the actual goal. The robot replans whenever it receives a sensor update and an updated $\overline{\mathcal{X}}_{t+1}^{\text{free}}$ from its non-deterministic filter, and accounts for ICS constraints [50] in-between sensor updates.

# D Proof of Proposition 2:

As shown in Proposition 1, the misdetection rate of the calibrated perception system $\bar{\phi}$ is $\epsilon$-bounded on environments drawn from $\mathcal{D}$ at each time step $t$, where the robot is at state $s_t$. In other words, the predicted occupied space $\hat{\mathcal{X}}_t^{\text{occ}}$ at each time step contains the true obstacles $A$ with high probability across environments. Conversely, the predicted free space $\hat{\mathcal{X}}_t^{\text{free}}$ at each time step does not contain the true obstacles $A$ with high probability across environments. If we consider a safety-relevant

misdetection cost at time step $t$:

$$\hat{C}_E^{\text{safe}}(\bar{\phi}, s_t) = \begin{cases} 1 & \text{if } A \subseteq \hat{\mathcal{X}}_t^{\text{free}} \text{ (unsafe)}, \\ 0 & \text{otherwise}, \end{cases} \tag{7}$$

then the misdetection rate over the set of states should be $\epsilon$-bounded across environments by Proposition 1:

$$\mathop{\mathbb{E}}_{E \sim \mathcal{D}_\mathcal{E}} \max_{t \in [0,T]} \hat{C}_E^{\text{safe}}(\bar{\phi}, s_t) \leq \epsilon. \tag{8}$$

Because the expectation in Equation (8) is over the set of environments, the following statement holds in any new environment (with probability $1 - \delta$ over the calibration dataset of environments),

$$\Pr\left\{ \max_{t \in [0,T]} \hat{C}_E^{\text{safe}}(\bar{\phi}, s_t) = 0 \right\} \geq 1 - \epsilon. \tag{9}$$

Given $m_t = \{\overline{\mathcal{X}}^{\text{free}}, \overline{\mathcal{X}}^{\text{occ}}, \overline{\mathcal{X}}^{\text{unknown}}\}$, a safe planner never drives the robot outside of the free space. Therefore, the safe planner guarantees $C_E^{\text{safe}}(\pi^{\bar{\phi}}) \leq \bar{C}_E^{\text{safe}}(\bar{\phi})$.

$$\Pr\left\{ C_E^{\text{safe}}(\pi^{\bar{\phi}}) = 0 \right\} \geq 1 - \epsilon. \tag{10}$$

# E    Extensions

In this section, we outline a few extensions to the basic technical approach described in Sections 4 and 5: (i) fine-tuning a pre-trained perception model, (ii) incorporating sensor and dynamics uncertainty, and (iii) calibrating perception modules beyond bounding box prediction.

## E.1    Fine-Tuning a Pre-Trained Perception Model

In Section 4, we assumed access to a pre-trained perception model $\phi$ that outputs bounding boxes. The conformal prediction-based uncertainty quantification procedure then uses the calibration dataset $D = \{E_1, \ldots, E_N\}$ of environments to produce a calibrated perception system $\bar{\phi}$ which lightly processes the outputs of $\phi$ by inflating the predicted bounding boxes. In practice, it may also be useful to *fine-tune* $\phi$ for our target deployment environments before performing uncertainty quantification.

This can be achieved using *split conformal prediction* [8], where one splits the overall dataset $D$ into $D = D_{\text{tune}} \cup D_{\text{cal}}$. If the perception model takes the form of a neural network $\phi_w$ parameterized by weights $w$, we can use $D_{\text{tune}}$ to fine-tune $w$ (or the weights of a residual network). We can then utilize $D_{\text{cal}}$ in order to perform the CP-based calibration as described in Section 4. As we demonstrate in Section 6, this additional fine-tuning step before calibration can reduce the conservatism of outputs and improve end-to-end success rates.

The typical choice of loss function for training a bounding box predictor is the *generalized intersection-over-union (gIoU) loss* [51]. This is a differentiable version of the IoU loss: given a ground-truth bounding box $A$ and a predicted box $B$, one computes $L(A, B) := |A \cap B|/|A \cup B|$. However, while this loss is popular in computer vision, it is not suitable for robot navigation. In particular, the IoU loss is *symmetric*: it does not distinguish between the ground-truth and predicted bounding box and thus does not encourage the predicted box to *contain* the ground-truth box. We propose a modification to the gIoU loss in Appendix E.1.1, which encourages that the predicted bounding box encloses the ground-truth box while also ensuring that the predicted box is not too large. Similar to the gIoU loss, this loss is (almost-everywhere) differentiable and scale invariant. We utilize this loss for fine-tuning in our experiments (Section 6). However, one could use any other method for finetuning not limited to training a simple neural network with gIoU loss [52].

### E.1.1    Loss Function for Fine-Tuning

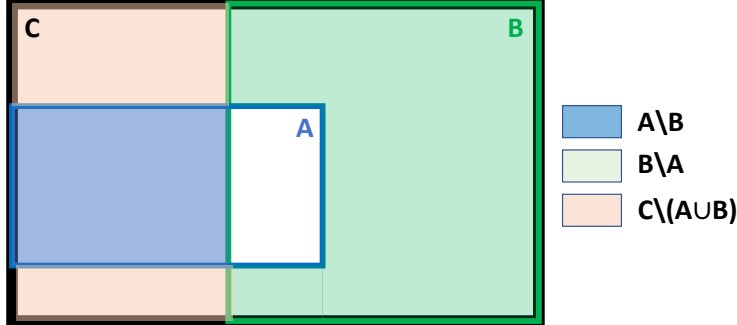

Figure 8: Visualization of different terms in the loss function for a single object setting.

We use a (almost-everywhere) differentiable loss function for training. The loss function seeks to ensure that the predicted shape (e.g., bounding box) encloses the ground truth shape while also ensuring that the predicted shape is not too large.

Let's consider the simplest setting wherein we have one object in the scene and we are making a single prediction. In this case, $A$ denotes the (convex) ground-truth shape and $B$ denotes the (convex) predicted shape. Let $C$ denote the convex hull of $A$ and $B$. Our loss function is a weighted combination of three terms,

$$L := w_1 l_1 + w_2 l_2 + w_3 l_3 = w_1 \frac{|A \backslash B|}{|A|} + w_2 \frac{|B \backslash A|}{|B|} + w_3 \frac{|C \backslash (A \cup B)|}{|C|}.$$

The first term is the most important; it tries to ensure that $B$ encloses $A$. The second term tries to make sure that $B$ is not much larger than it needs to be, see Figure 8. The first and second terms are sufficient if $A$ and $B$ are overlapping. However, if they do not overlap, there is no gradient information provided by the first two terms. Following [51], we introduce a third loss term in order to provide gradient information when the shapes do not intersect. The loss terms $l_1, l_2, l_3$ are each bounded within $[0, 1]$. Hence, if we choose $w_1, w_2, w_3$ such that $\sum_i w_i = 1$, then the overall loss is also bounded within $[0, 1]$. Now let's consider the setting wherein, $A$ denotes the union of multiple ground-truth bounding boxes (say we have $m$ objects in the scene) and $B$ is the union of all the predicted bounding boxes (we predict $n$ boxes). We consider all the individual bounding box predictions $B_i, \forall i \in \{1, \ldots n\}$ and associate the closest *visible* ground-truth bounding box $A_i$ to each prediction. Now we can define $C_i$ as the convex hull of $A_i$ and $B_i$ and the resulting loss function, $L_i$,

$$L_i := w_1 \frac{|A_i \backslash B_i|}{|A_i|} + w_2 \frac{|B_i \backslash A_i|}{|B_i|} + w_3 \frac{|C_i \backslash (A_i \cup B_i)|}{|C_i|}.$$

Hence, the overall loss is,

$$L = \frac{1}{n} \sum_{i=1}^{n} L_i.$$

Please refer to [51, Appendix 4.3] for instructions on how to compute the loss analytically for axis-aligned bounding boxes.

### E.1.2 Simulation Results - Effect of finetuning dataset size

Upon collecting a calibration dataset of $\sim 400$ environments, as described in the experiment setup in Section 6, we may choose to use a smaller subset of the calibration dataset to further finetune the pre-trained perception model to perform better in the types of environments we are interested in deploying the robot in. We consider the effect of different dataset split sizes for finetuning and then calibration. Using a larger set of environments for finetuning $|D_{\text{tune}}|$ may result in a better tuned model, but will leave fewer environments for calibration, $|D_{\text{cal}}|$, resulting in a more conservative $\hat{\epsilon}$ and $\hat{q}_{1-\epsilon}$ that satisfies the dataset-conditional guarantee (3), and vice versa. This trade-off is seen

in Table 1, where we observe the best performance when we have an equal split between finetuning and calibration.

| Split size ( $|D_{\textbf{tune}}| + |D_{\textbf{cal}}|$ ) | $\hat{q}_{0.85}$ (in m) | Collision | Misdetection | Goal Reached |
|---|---|---|---|---|
| $100 + 300$ | 0.68 | 0% | 1% | 89% |
| $200 + 200$ | **0.64** | 0% | **1%** | **94%** |
| $300 + 100$ | 0.93 | 0% | 2% | 76% |

Table 1: A comparison of the effect of various partition sizes for finetuning and calibration for PwC.

### E.2 Sensor Errors and Dynamics Uncertainty

In Section 2, we modeled the robot's sensor as a deterministic mapping $\sigma : \mathcal{S} \times \mathcal{E} \to \mathcal{O}$, which provides observations from a particular state in a given environment. This formulation allows us to also incorporate sensor errors. Specifically, any errors or randomness in the sensor can be formally included as part of the environment $E \in \mathcal{E}$. Thus, in addition to sampling environmental variables such as obstacle locations, geometries, etc., each environment $E$ also samples random variables that prescribe sensor errors from each state $s \in \mathcal{S}$ in the environment. This way of modeling sensor errors allows $\sigma$ to be deterministic (since all sources of randomness are included in $E$), allows the sensor errors to be dependent on the relative pose of the robot relative to obstacles (e.g., modeling the fact that depth estimates are often further from ground-truth depth values as distance increases), and also allows us to model correlations in sensor errors from different locations (e.g., capturing the fact that sensor errors from nearby robot locations can be highly correlated). Modeling time-varying sensor errors (i.e., different sensor errors from the robot state at different times) is not as immediate, but could potentially be incorporated by augmenting the state space $\mathcal{S}$ to include the time-step.

In addition to errors in sensing, one can also account for uncertainty in the dynamics of the robot by using a robust planner (see [16] for an overview). In the experiments described in Section 7, we incorporate uncertainty by generating plans that prevent the robot from entering the inevitable collision set (cf. Section 5) even with bounded uncertainty in the dynamics.

### E.3 Calibration with General Occupancy Prediction Models

Section 4 introduced the CP-based calibration procedure in the context of bounding box prediction. However, the theoretical formulation in Section 4 is applicable to more general occupancy prediction models; the key requirement is the presence of a scalar quantity that monotonically grows the size of the predicted occupied space (e.g., the inflation parameter $q$ for bounding boxes in Section 4). This allows one to define the non-conformity score $U_i$ for an environment $E_i$ as in (4) to be the smallest scalar such that the inflated predicted occupied space contains the ground-truth obstacles (for all robot locations). Hence, we can calibrate the outputs of any perception system that predicts an occupied set or performs *occupancy prediction* more generally, i.e., assigns a (heuristic) occupancy confidence to each point in the space. Possibilities for the latter include scene completion networks [47] or deep signed-distance function representations [53]. A threshold on this confidence acts as the scalar parameter that monotonically controls the size of the predicted occupied space. The conformal prediction procedure from Section 4 can then be used to find a confidence threshold such that predicted occupied space contains the true occupied space (with probability $1 - \epsilon$ in a new environment).

## F   Calibration and planning

 We collect a calibration dataset of 400 environments wherein we randomly place $1 - 5$ chairs from the diverse 3D-Front dataset [11] in a 8 m $\times$8 m room. In this 8 m $\times$8 m space, we use a fixed set of 2000 sampled configurations for the sampling-based motion planner and use the same set of samples for the calibration procedure. We construct the calibration dataset in simulation using CAD models of *real* furniture pieces from the 3D-Front dataset [11], which contains a highly diverse array of industrial CAD models developed by professional designers to ensure that the performance of the

perception system remains the same in its simulation and hardware implementation. Similarly, we collect an additional fine-tuning dataset $D_{\text{tune}}$ consisting of 100 environments. These environments include ones with occlusions of the goal and objects in the scene.

## F.1 Metrics for experiments

We simulate the dynamics of the Unitree Go1 quadruped robot and task the robot with navigating to a goal location that is $\sim 7$m away from the initial location of the robot. The robot camera has a field of view of $70°$ and a visibility range of $[1,5]$ m. With an allowable misdetection rate of $\epsilon = 0.15$, we obtain $\hat{q}_{0.85} = 0.75$ m for PwC, $\hat{q}_{0.85} = 0.65$ m for PwC-fine-tuned, and $\hat{q}_{0.85} = 0.05$ m for CP-avg. through calibration. The planner replans and obtains a new sensor observation to update the filter every $0.5$ s or less (if the previous plan is already completed).

We utilize the following metrics for our simulation experiments: a trial is counted as a collision if the robot collides with an obstacle and we count a misdetection for a trial if the free space predicted by the planner has any intersection with the ground-truth bounding boxes of the obstacles. We say that the goal has been reached in a given trial if the robot is able to navigate to within 1 m around the goal in less than 140 s. We also record the average path length for trials in which the goal is reached.

## F.2 Results: Effects of varying $\epsilon$ on safety rate

We compare our method, PwC, to the baseline CP-avg. We vary the allowable safety rate $\epsilon$ for each method, and compute the rate of safety in 100 test environments. As seen in Table 2 and Figure 5, our method guarantees not only that the rate of misdetections are guaranteed to be bounded, but also the safety rate. The safety rate of PwC is also consistently better than that of CP-avg.

| $\epsilon$ | CP-avg. | PwC |
|---|---|---|
| 0.20 | 95% | **100%** |
| 0.10 | 98% | **99%** |
| 0.15 | 99% | **100%** |
| 0.10 | 98% | **100%** |
| 0.05 | 98% | **100%** |

Table 2: A comparison of the safety rates of CP-avg. and PwC when we vary the confidence threshold $\epsilon$.

## F.3 Results: Effect of varying the number of sampled configurations

For our experiments, we used a fixed set of 2000 sampled configurations. However, depending on the planner configuration requirements and desired speed of computation, the user may decide to have a different number of configuration samples for calibration and planning. We study the change in the CP inflation, $\hat{q}_{0.85}$, the resulting collision, misdetection, and task completion (reaching goal) rates. As we can see in Table 3, in our case, we have far fewer misdetections with fewer samples, but we also observe a decrease in number of times the robot reaches the goal. We suspect that with fewer samples of configurations (consisting of $x, y, v_x, v_y$), it is harder for the sampling-based motion planner to find feasible paths. On the other hand, we also observe a less conservative $\hat{q}_{0.85}$ when we use fewer samples; this is presumably also a result of using fewer samples to compute the non-conformity score that comprises of the worst-case perception error across all configurations.

| # samples | $\hat{q}_{0.85}$ | Collision % | Misdetection % | Goal Reached % |
|---|---|---|---|---|
| 1050 | 0.7086 | 0% | 1% | 47% |
| 1500 | **0.6910** | 0% | **0%** | 57% |
| 2000 | 0.75 | 0% | 7% | **90%** |

Table 3: A comparison of the CP inflation, $\hat{q}_{0.85}$ when we vary the number of sampled configurations.

## F.4 Results: Effects of closed-loop distribution shift on misdetections

In addition to the challenge of generalization, we highlight another challenge that any uncertainty quantification method for perception must tackle. Suppose we fix a policy $\pi^\phi$ (that uses perception system $\phi$) and collect a dataset of observations in different calibration environments from the states that result from applying $\pi^\phi$. We can use ground-truth bounding boxes in these environments to produce a calibrated perception system $\bar{\phi}$ with a statistical assurance on correctness for the distribution of observations induced by $\pi^\phi$. However, if

| Method | Collision | Mis-detection | KL-divergence |
|---|---|---|---|
| CP-avg. ($w = 1$) | 14% | 54% | 2.09 |
| CP-avg. ($w = 10$) | 2% | 64% | 2.72 |
| PwC ($w = 1$) | 0% | 0% | 1.48 |
| PwC ($w = 10$) | 0% | 2% | 2.04 |

Table 4: A comparison of the effect of changing the planner parameters on CP-avg. and PwC.

we now apply the policy $\pi^{\bar{\phi}}$ using the *calibrated* perception system $\bar{\phi}$, the resulting distribution of states will be *different* from the distribution that forms the calibration dataset, thus invalidating the statistical assurance. We refer to this challenge as *closed-loop distribution shift*, which is similar to challenges that arise in offline reinforcement learning [54] and imitation learning [55].

To illustrate the effect of closed-loop distribution shifts on misdetections, we used exactly the same setup described above to obtain the simulation results in Figure 4. We changed the planner cost to have a different weighting on the cost-to-go. For one setting, we chose a weight $w = 1$ on the cost-to-go, which is the same as the weighting on the cost-to-come. In another setting, we chose a weight $w = 10$ on the cost-to-go, and hence a $10\times$ more emphasis on the cost-to-go compared to the cost-to-come. Table 4 shows the KL-divergence between the states visited by the planner and the sampling distribution of states for calibration as a measure of the closed-loop distribution shift. Increasing closed-loop shifts lead to higher misdetections. One can see that a simple change in the planner parameters can lead to potentially large changes in the safety rates for CP-avg. The closed-loop shift we may see in practice is unknown apriori. Hence, it is difficult to make any statements on the planner safety in closed-loop despite using CP for calibration of the perception system. PwC, on the other hand, is robust to the closed-loop shifts and can still satisfy the misdetection and safety assurance regardless of the planner parameters used.

### F.5   Results: Comparison to heuristic inflation

We compare PwC to the baseline method of inflating the bounding box predictions based on some heuristic confidence level, i.e., we scale the bounding box with 1 - confidence (so we scale the boxes where we are less confident by a larger amount). While this baseline

| Method | Collision % | Misdetection % | Goal Reached % |
|--------|-------------|----------------|----------------|
| PwC | **0**% | **7**% | 90% |
| Heuristic | 3% | 67% | **97**% |

Table 5: A comparison of the effects of using heuristic inflation versus PwC.

demonstrates a higher completion rate, both the collision rate and the misdetection rate increase significantly, leading to unsafe situations. Further, while our method provides a statistical safety guarantee, the baseline method does not admit any formal assurance.

## G   Hardware

### G.1   Hardware and Environmental Setup

We represent the robot's state as $s_t = [x, y, v_x, v_y]^T$ where $x$ and $y$ are its position in the environment and $v_x$ and $v_y$ are the respective velocities. For each trial, the robot is initialized around position $[4, 0]$m (with the origin set to bottom left corner of the room) and has a time horizon of 60 seconds to reach the goal within a 1m radius. The robot replans every 1s in a receding horizon manner using the safe planner described in Section 5. The goals are varied every 10 environments and include positions $[2, 7]$m, $[7, 0]$m, and $[6, 7]$m.

**Hardware.** We use the Unitree Go1 quadruped robot with fully onboard sensing and computation. The robot is equipped with a ZED 2i RGB-D camera and a ZED Box computer attached to the base of the robot as shown in the top row of Figure 7. The Zed 2i provides the Go1 with point cloud observations with a 70° field of view and a visibility range of $[1, 5]$m. The Zed 2i also uses vision-inertial odometry to provide accurate positional state estimates in the environment. The Zed Box includes an 8-core ARM processor and a 16GB Orin NX GPU. This allows us to process the point cloud observations in order to produce bounding boxes using the pre-trained 3DETR model [20]. The bounding boxes are aggregated over time to update the estimated free, occupied, and unknown spaces as described in Section 5. The safe planner described in Section 5 is used to output Cartesian velocity commands bounded at a speed of 0.8m/s; these commands are sent from the Zed Box over UDP to the Go1's processor. Our method is implemented real-time on the Zed Box hardware with replanning every 0.5 seconds of which the non-deterministic filter takes 0.00025 seconds to run. The dynamics of the Go1 are estimated using MATLAB's System Identification Toolbox [56] and are provided in Appendix G.2.

**Environments.** We test the robot in 30 different environments, consisting of various chair configurations and geometries in an 8 m ×8 m room. Configurations range from random, occluded goal, occluded chairs, clustered chairs, and narrow paths (approximately 1.8m in width leaving 0.4m of available freespace for PwC to find). Each environment has between 4 and 8 chairs present. See Appendix G.3 and G.4 for the unseen chairs used in testing and the environment configurations respectively. We use a Vicon motion capture system to log the ground-truth placement and bounding boxes of the chairs for each environment.

### G.2 System Identification

To perform system identification of the Unitree Go1 quadruped robot, we collected trajectories using a Vicon motion capture system. We then used MATLAB's system identification toolbox [56]. Specifically, we provided an initial linear ODE grey box model guess and then used prediction error minimization (PEM) for refinement. The resulting system is shown in (11) where $x$ and $y$ describe the positional state of the robot in the environment, $v_x$ and $v_y$ describe the respective velocities, and $u_x$ and $u_y$ describe the respective commanded velocities.

$$\begin{bmatrix} \dot{x} \\ \dot{y} \\ \dot{v}_x \\ \dot{v}_y \end{bmatrix} = \begin{bmatrix} 0 & 0 & 1 & 0 \\ 0 & 0 & 0 & 1 \\ 0 & 0 & -2.5170 & 0.1353 \\ 0 & 0 & -0.5197 & -3.9680 \end{bmatrix} \begin{bmatrix} x \\ y \\ v_x \\ v_y \end{bmatrix} + \begin{bmatrix} 0 & 0 \\ 0 & 0 \\ 2.3350 & 0 \\ 0 & 4.6510 \end{bmatrix} \begin{bmatrix} u_x \\ u_y \end{bmatrix} \tag{11}$$

### G.3 Chair Test Dataset

Our test dataset of chairs for the experiments conducted in Section 7 included 8 chairs with diverse sizes and geometries unseen in training and calibration for the perception system. Test chairs are shown below in Figure 9.

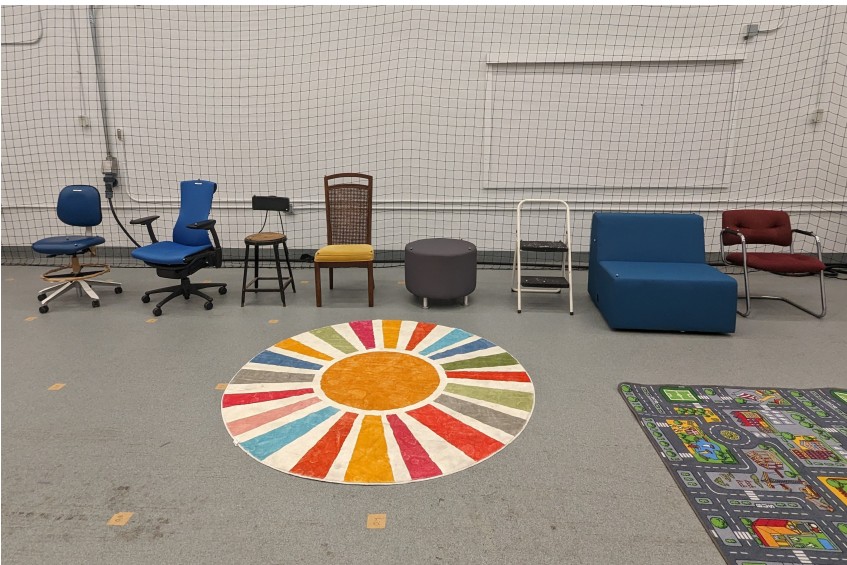

Figure 9: New, unseen test chairs used in hardware experiments.

### G.4 Environments

As described in Section 7, the robot was tested in 30 unique environments with varying furniture configurations and goals. The following 30 figures show an image of each configuration, accompanied by a bird's-eye map of the obstacle and goal locations.

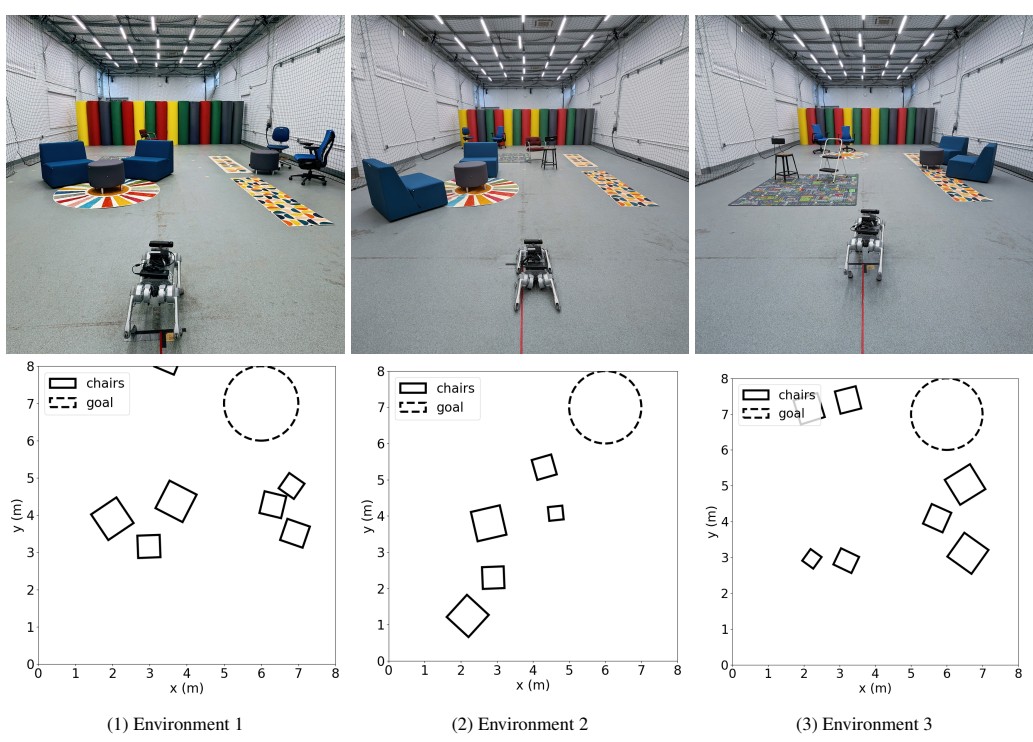

(1) Environment 1      (2) Environment 2      (3) Environment 3

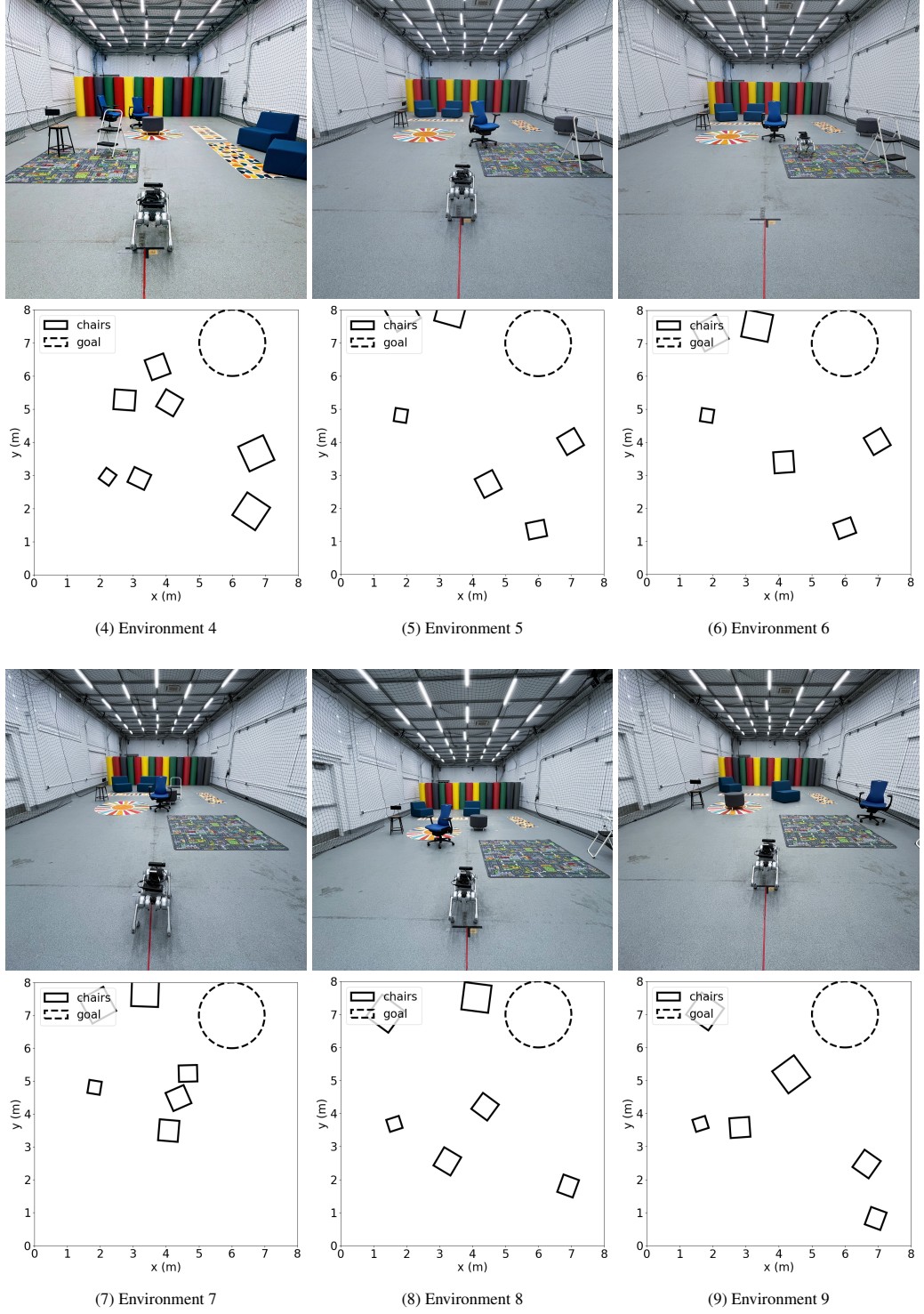

(4) Environment 4

(5) Environment 5

(6) Environment 6

(7) Environment 7

(8) Environment 8

(9) Environment 9

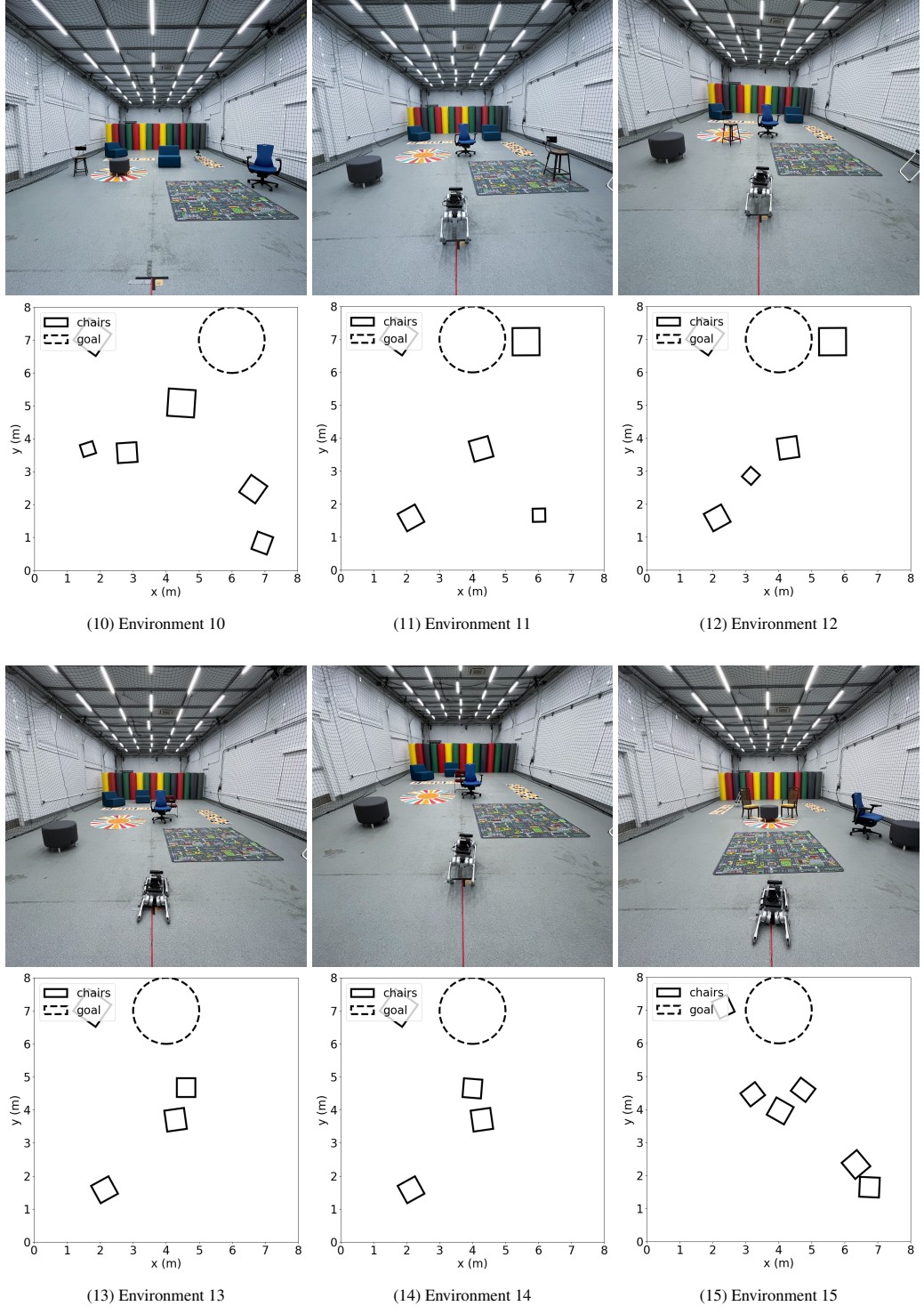

(10) Environment 10      (11) Environment 11      (12) Environment 12

(13) Environment 13      (14) Environment 14      (15) Environment 15

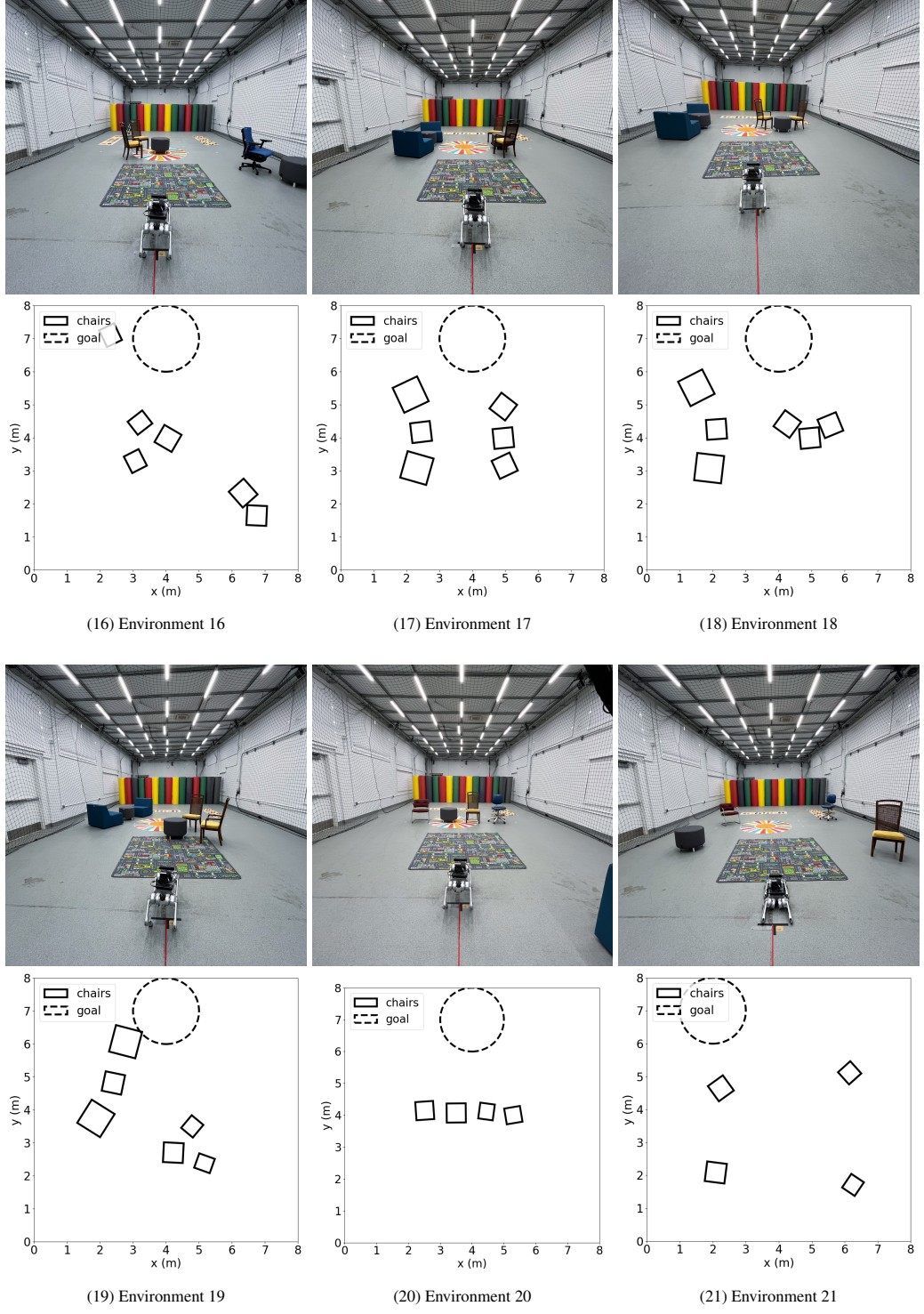

(16) Environment 16      (17) Environment 17      (18) Environment 18

(19) Environment 19      (20) Environment 20      (21) Environment 21

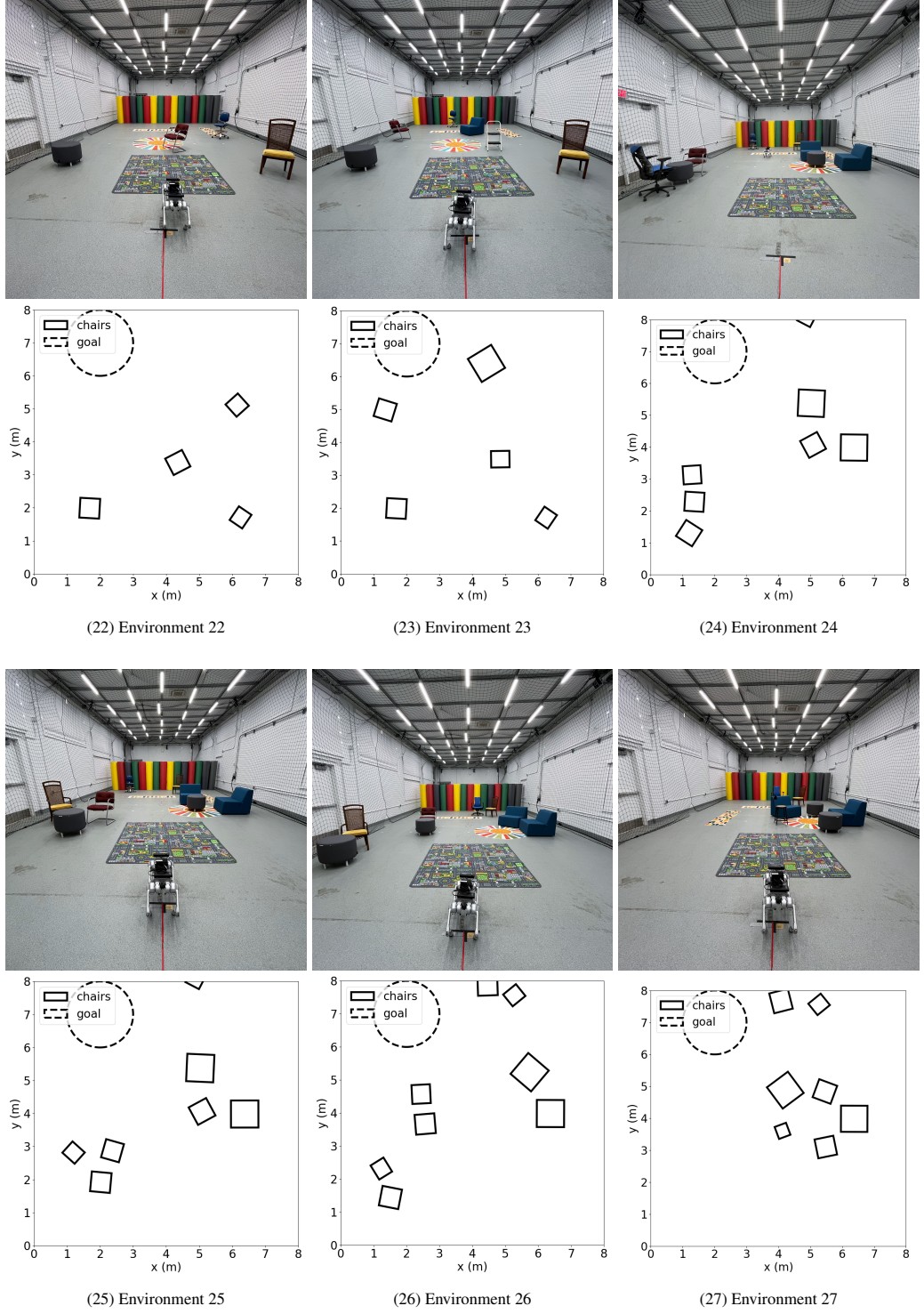

(22) Environment 22          (23) Environment 23          (24) Environment 24

(25) Environment 25          (26) Environment 26          (27) Environment 27

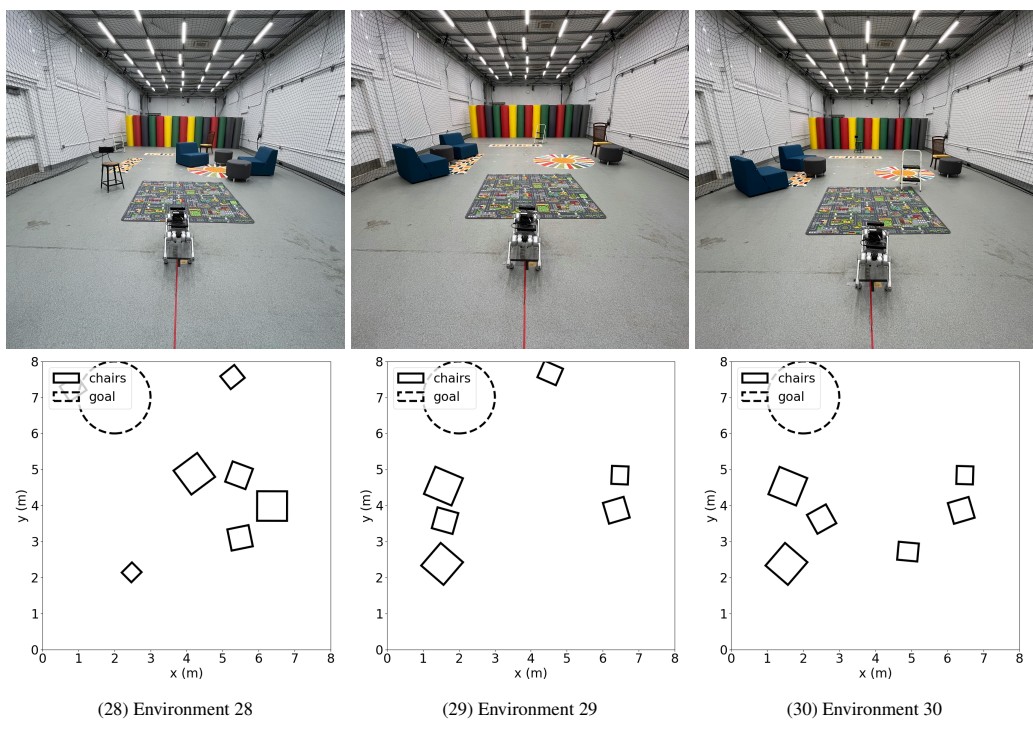

(28) Environment 28          (29) Environment 29          (30) Environment 30

