# OpenReview forum: "Perceive With Confidence: Statistical Safety Assurances for Navigation with Learning-Based Perception"
_robot-learning.org/CoRL/2024/Conference — CoRL 2024_

### Official Review · Reviewer_B6rP · 2024-07-17
**Review of "Perceive With Confidence: Statistical Safety Assurances for Navigation with Learning-Based Perception"**

**Originality:** 3
**Technical Quality:** 3
**Clarity Of Presentation:** 2
**Potential Impact:** 3
**Recommendation:** 3
**Confidence:** 4

**Review:**

The paper is well-structured and provides a thorough investigation into the problem of ensuring safety in navigation with learning-based perception. The methodology is sound and well-explained, making a significant contribution to the field of robotics. The writing is clear and concise, making complex concepts accessible. The approach of using conformal prediction for calibrating perception systems to handle distribution shifts is novel and addresses a crucial gap in the current literature. This work is the first to provide such end-to-end safety assurances while being robust to closed-loop distribution shifts. The proposed framework has significant implications for the deployment of robots in real-world environments.

Cons:

 • The framework assumes static obstacles in the environment, which limits its applicability to more dynamic, real-world scenarios where obstacles may move unpredictably. This assumption might hinder the deployment of the system in environments such as urban settings or areas with moving people and vehicles.

 • The integration of the non-deterministic filter and safe planner adds computational overhead, which might affect real-time performance. The paper does not provide detailed benchmarks on the real-time efficiency of the system, particularly under high-speed navigation scenarios or in environments requiring quick decision-making.

**Quality Of The Limitations Section:**

2

**Questions For Rebuttal:**

• The use of bounding box inflation to ensure safety can introduce conservatism, potentially affecting the robot’s efficiency. Can you discuss potential strategies to mitigate this conservatism without compromising safety?

 • The results indicate that varying the confidence threshold (ϵ) affects the misdetection rate. Can you provide more insights into how different safety thresholds impact overall system performance, and what guidelines would you suggest for selecting an appropriate threshold?

**Robotics Focus:**

4

**Summary Of Paper:**

This paper presents a novel framework, Perceive With Confidence (PWC), that provides statistical safety assurances for navigation in robots using learning-based perception models. By leveraging conformal prediction for uncertainty quantification, the authors ensure robustness to distribution shifts in environments, leading to improved safety in both simulated and real-world scenarios.

**Summary Of Recommendation:**

The proposed framework, Perceive With Confidence (PWC), introduces a novel approach to uncertainty quantification through conformal prediction, providing robustness to distribution shifts. The thorough validation through both simulation and hardware experiments demonstrates the practicality and effectiveness of the approach. Despite some limitations, such as the assumption of static obstacles and the conservatism introduced by bounding box inflation, the paper offers substantial advancements in the field of robotics and presents a solid foundation for future research.

---

### Official Review · Reviewer_a8KQ · 2024-07-20
**A promising uncertainty calibration framework for safe navigation with learning-based perception**

**Originality:** 3
**Technical Quality:** 3
**Clarity Of Presentation:** 2
**Potential Impact:** 3
**Recommendation:** 3
**Confidence:** 3

**Review:**

### Strengths
- The proposed idea is neat and promising in facilitating safe planning while exploiting the pre-trained learning-based perception models.
- The target problem is highly relevant to the robotic perception, planning, and control communities, facilitating the co-development of an integrated closed-loop autonomous system.
- The paper is overall good to follow with a clear problem formulation and experiments demonstration except for the description of the method part (detailed below).
- Technically, the idea of ensuring statistical guarantee for outputs of a learning-based model via conformal prediction is not new. Though, exploiting such formal assurance for safe planning highlights the originality of this work. The derived assurance for a safe policy induced by the calibrated perception model in proposition 2 corroborates this aspect.
- The benefits of the proposed idea are verified in both simulation and the real world in terms of comprehensive metrics such as collision rate, mis-detection rate, success rate, etc.  The results of the proportional relation between the specified confidence thresholds and the actual mis-detection rate further confirm the theoretical statements in the main paper.

### Weaknesses
- For the presentation of the method part:
   1. Lack of brief preliminaries on the core tool used by this work, i.e., conformal prediction in the **main** text, which confuses the reader when evaluating the novelty of the proposed method;
   2. In Section 3.1, it is confusing to highlight only the "closed-loop distribution shift" in bold but not the "generalization." Moreover, it would be preferred if there is a clear definition of each term to replace the a bit wordy description with a large paragraph for conciseness;
   3. It is a bit awkward for the readers to understand Equation 3. without a proper introduction (maybe this part can be added along with the background section);
   4. Regarding the explanation of the non-deterministic filter, a formal description would be highly recommended for readability. The current explanation is done through a long paragraph and the low-resolution Figure 2., which might be time-consuming for the reader to understand the core idea easily;
- For the technical part, it is hard to find Equation 3 in [2], as mentioned by the authors in the appendix, which should be the key difference to the version of conformal prediction used mostly from my understanding. A more detailed and self-contained description of this part is better to have for convincing the readers who are interested in getting familiar with the technical details;
- Moreover, it seems less convincing not to discuss the covariate shift and general distribution shift for conformal prediction [1]. From my understanding, the so-called "closed-loop distribution shift" is one instance of the covariate shift [1]. I see little value in creating a new term, as it can easily distract readers with such a background. In this sense, if the weighted conformal prediction [1] is one of the principled approaches to address this problem, it would be better to discuss or even include it as a baseline in the experiments;
- On the other hand, empirically, comparing the uncertainty-aware or OOD-aware perception models such as [3] would be sensible to discuss or be added as a baseline in the experiments. Even though this type of method provides only heuristic-based uncertainty quantification, in practice, the implementation is quite similar, i.e., thresholding. The major limitation is the lack of formal assurance, which is addressed by the proposed approach. Comparing them can also hint at whether formal assurance indeed works better for perception-based closed-loop planning. Considering this, adding this part would be, in general, of great interest to the community of uncertainty quantification in robotics.
- Lack of an ablation study of the number of samples for approximating "all" states during calibration: In the footnote on page 3, the "all" possible states are approximated with a fixed set of samples in the planner.
- Lack of discussion of the sim-to-real gap in using the perception module calibrated in simulation to the real world directly;
- Last but not least, in the experiments, it's pretty nice to see the empirical validation of the derived proposition 1 in Figure 5. Would it be more appealing and convincing if a similar trend can be observed for the assured safety cost stated in proposition 2;


[1] Barber, Rina Foygel, et al. "Conformal prediction beyond exchangeability." The Annals of Statistics 51.2 (2023): 816-845.

[2] Vovk, Vladimir. "Conditional validity of inductive conformal predictors." Asian conference on machine learning. PMLR, 2012.

[3] Feng, Jianxiang, et al. "Topology-Matching Normalizing Flows for Out-of-Distribution Detection in Robot Learning." 7th Annual Conference on Robot Learning.

**Quality Of The Limitations Section:**

3

**Questions For Rebuttal:**

My suggestion is to address the points articulated in the list of weakness above.

**Robotics Focus:**

4

**Summary Of Paper:**

To handle the reliability issue in using pre-trained learning-based perception models for planning in navigation, the authors proposed a simple yet effective uncertainty quantication framework based on conformal prediction. The proposed method can provide formal assurance of the correctness in its outputs and is robust against the state distribution shift during closed-loop deployment. Emprical evaluation of the proposed idea has been conducted in both simulation and real-world, showing strong benefits in decreasing collision and mis-detection rates with comparable success rates.

**Summary Of Recommendation:**

Due to the insufficient clarity in the presentation and lack of dicussion/experiments for the related work, I would vote for a weak reject at the current stage.

---

### Official Review · Reviewer_UaeJ · 2024-07-23
**The paper addresses an important topic of safety of perception-based systems. While interesting, some technical concerns need to be addressed.**

**Originality:** 3
**Technical Quality:** 4
**Clarity Of Presentation:** 4
**Potential Impact:** 3
**Recommendation:** 2
**Confidence:** 4

**Review:**

- Clarity: The paper is well-written and easy to follow.

- Originality: The use of Conformal Prediction (CP) has recently gained much interest. The paper follows a simple CP argument to provide some notion of the correctness of pre-trained models. Nevertheless, the use of CP in this paper needs to be more carefully elaborated to avoid the common pitfalls in using CP (e.g., hidden assumptions on the model and the distribution).

- Significance: Appropriate.

- Strengths:
The paper provides a comprehensive evaluation of the benefits of using the proposed method.

- Weaknesses:

- - (1) The definition of "environment" is very simplistic. In this paper, the environment affects the obstacles' location and geometry. What about the remaining factors like lighting conditions, weather, ... etc. In practice, one can not get an i.i.d sample that efficiently covers all these environmental changes, which strictly limits the applicability of the proposed methodology.

- - (2) One of the major weaknesses of vision-based models is that ``they can not always detect objects from all angles and under all environmental conditions.'' In other words, vision-based models frequently may not return any bounding box, although the obstacle is in the scene (e.g., what happened in the infamous Uber accident in Arizona). It is not clear how Proposition 1 captures such a case. The current proposition (as it is currently presented) seems to be implicitly assuming that the object detector can always detect the objects but only the bounding box is what needs to be calibrated, which is a very strong assumption given the current state of vision models.

- - (3) The footnote 1 (on Page 3) is vague. Do you mean that one has to fix a set of states to be used during the calibration and use the same set of states to be used by the online motion planner? If this is the case, such a (strong) assumption needs to be explicitly stated in Proposition 2, as motion planners who do not use this set of states during online planning are no longer safe.

**Quality Of The Limitations Section:**

2

**Questions For Rebuttal:**

The authors should address the weaknesses number (1), (2), and (3) in the review above.

**Robotics Focus:**

4

**Summary Of Paper:**

The paper proposes the use of Conformal Prediction (CP) to calibrate the output of object detectors. The paper uses standard CP argument to show statistical guarantees of the classification model. The paper demonstrates the efficiency of the proposed method using multiple experiments.

**Summary Of Recommendation:**

While the paper is interesting, the technical argument behind Proposition 1 and 2 is not thorough and several hidden (and strong) assumptions are not thought out thoroughly.

---

### Author Rebuttal · Authors · 2024-08-07

We thank the reviewers and the area chair for their thoughtful reviews that have helped improve the paper. We address all reviewer comments individually below and provide a revised version of the paper including the feedback from the reviewers. The changes in the revision are typeset in blue.

---

### Decision · Program_Chairs · 2024-09-04

**Decision:**

Accept

**Comment:**

The paper is an interesting application of CP to object detection. The reviewers bring up several issues that should be addressed in the rebuttal:

- It is unclear which states can be used by the motion planner, is it the same set of states used for calibration?
- The explanation of the non-deterministic filter is unclear.
- The paper should be more self-contained (hard to find Equation 3 in [2])
- The paper would benefit from comparisons to other baselines and ablations
- There needs to be discussion of sim2real for this kind of approach
- Need to discuss the speed of the method

Please see reviewer comments for more details.

-----Post-rebuttal comments----

The authors' rebuttal clarified several issues raised by the reviewers, but the reviewers' ratings remain unchanged. Overall, the need for safety guarantees on learned systems is quite critical, so this paper is in an area that should be given more attention. While there are some limitations to the method (e.g. the fixed set of samples), there is still a contribution here.